# Genetic Reconstruction and Forensic Analysis of Chinese Shandong and Yunnan Han Populations by Co-Analyzing Y Chromosomal STRs and SNPs

**DOI:** 10.3390/genes11070743

**Published:** 2020-07-03

**Authors:** Caiyong Yin, Kaiyuan Su, Ziwei He, Dian Zhai, Kejian Guo, Xueyun Chen, Li Jin, Shilin Li

**Affiliations:** 1Department of Anthropology and Human Genetics, School of Life Sciences, Fudan University, Shanghai 200438, China; cyyin18@fudan.edu.cn (C.Y.); 16110700051@fudan.edu.cn (K.S.); heziwei@fudanjushe.com (Z.H.); 19260700012@fudan.edu.cn (K.G.); 2Human Phenome Institute, Fudan University, Shanghai 200438, China; 3Criminal Investigation Department of Yunnan Province, Kunming 650021, China; zd3537cn@aliyun.com (D.Z.); 15887818177@163.com (X.C.); 4Public Security Bureau of Zibo City, Zibo 255000, China; 5State Key Laboratory of Genetic Engineering, Collaborative Innovation Center for Genetics and Development, School of Life Sciences, Fudan University, Shanghai 200438, China; 6Department of Forensic Medicine, School of Basic Medical Sciences, Fudan University, Shanghai 200032, China

**Keywords:** population investigation, Y-STR, Y-SNP, population differentiation

## Abstract

Y chromosomal short tandem repeats (Y-STRs) have been widely harnessed for forensic applications, such as pedigree source searching from public security databases and male identification from male–female mixed samples. For various populations, databases composed of Y-STR haplotypes have been built to provide investigating leads for solving difficult or cold cases. Recently, the supplementary application of Y chromosomal haplogroup-determining single-nucleotide polymorphisms (SNPs) for forensic purposes was under heated debate. This study provides Y-STR haplotypes for 27 markers typed by the Yfiler^™^ Plus kit and Y-SNP haplogroups defined by 24 loci within the Y-SNP Pedigree Tagging System for Shandong Han (*n* = 305) and Yunnan Han (*n* = 565) populations. The genetic backgrounds of these two populations were explicitly characterized by the analysis of molecular variance (AMOVA) and multi-dimensional scaling (MDS) plots based on 27 Y-STRs. Then, population comparisons were conducted by observing Y-SNP allelic frequencies and Y-SNP haplogroups distribution, estimating forensic parameters, and depicting distribution spectrums of Y-STR alleles in sub-haplogroups. The Y-STR variants, including null alleles, intermedia alleles, and copy number variations (CNVs), were co-listed, and a strong correlation between Y-STR allele variants (“DYS518~.2” alleles) and the Y-SNP haplogroup QR-M45 was observed. A network was reconstructed to illustrate the evolutionary pathway and to figure out the ancestral mutation event. Also, a phylogenetic tree on the individual level was constructed to observe the relevance of the Y-STR haplotypes to the Y-SNP haplogroups. This study provides the evidence that basic genetic backgrounds, which were revealed by both Y-STR and Y-SNP loci, would be useful for uncovering detailed population differences and, more importantly, demonstrates the contributing role of Y-SNPs in population differentiation and male pedigree discrimination.

## 1. Introduction

Y chromosomal short tandem repeats (Y-STRs) refer to simple repeats of a 2–6 bp spreading across the Y chromosome and are a set of polymorphic genetic markers in linkage. Primarily, they are used in practical work for forensic purposes, such as paternity tests [1], criminal detections [2], and body identifications in natural disasters [3]. In previous studies, investigators found that Y-STR loci mutate fast through Y chromosome transmission (3.78 × 10^−4^ to 7.44 × 10^−2^ mutation/generation) [4]. Slowly mutating Y chromosomal single-nucleotide polymorphisms (Y-SNPs) are able to build stable phylogeny by defining stable haplotypes, termed haplogroups [5,6,7]. For forensic population genetics, adding of Y-SNPs into male population structure construction would allow a deeper understanding of population characteristics, population differentiation, and haplotype/gene diversities.

Among Chinese Han populations, huge efforts have been made to dig into population genetic backgrounds via genome-wide markers. Gao et al. developed a population genome database called “PGG.Han”, which enrolls over 114,000 Han individuals [8]. Previously, determined by the genetic make-up of various Han populations, Han populations were divided into northern Han, central Han, and southern Han [9], which is nearly consistent with their geographic locations in latitude. For forensic applications, population investigations by Y-STRs also generate much knowledge on inter-relationships among distinct male populations. In order to dissect the genetic structure of male populations, co-analysis of Y-STR and Y-SNP was conducted for diverse populations [10,11,12,13,14,15,16,17] but not for the majority of Chinese Han populations. In this research, preliminary comparative studies were conducted by genotyping commonly used Y-STRs and low-resolution Y-SNPs in two Chinese populations—Shandong Han and Yunnan Han—to characterize the patrilineal patterns in population genetics.

Located in the middle and lower reaches of the Yellow River, Shandong province is on the northeastern coast of China mainland. It is the birthplaces of Chinese culture and is of great importance to Taoism, Buddhism, and Confucianism in China. Shandong is the second most populous province in China. According to the 2010 China census (http://www.stats.gov.cn/tjsj/pcsj/rkpc/6rp/indexch.htm), the population size of Shandong is about 89 million, and Shandong population is mainly composed of the Han nationality (98.02%). As for Yunnan province, it is the most southwestern region in China and possesses the highest ethnic diversity in China (total population size ≈ 46 million). In addition to the Han nationality, there exist 25 diverse ethnic minorities with the population size greater than 6000, including Yi, Hani, Bai, Dai, Zhuang, and others. Among these, 15 are unique to Yunnan. Individuals belonging to minority groups account for 33.4%. Years of cultural exchanges and gene flows have resulted in mixed ethnic components in Yunnan Han population.

In this study, 24 Y haplogroup-determining SNP markers and 27 Y-STR loci were typed to co-analyze the basic genetic structure of the two populations. Phylogenetic trees were reconstructed by 27 Y-STR haplotypes on both population and individual scales. Variants including null alleles, intermedia alleles, and copy number variations were observed, among which all “.2” micro-variants at the DYS518 locus were identified within the QR haplogroup. Further, a median-joining network was constructed to illustrate the evolutionary pathway of these samples. Varying spectrums of various Y-STR alleles within the C2, O1, and O2 haplogroups were demonstrated between Shandong and Yunnan Han populations. The underlying genetic characteristics, as well as initiative Y chromosomal variations of Shandong and Yunnan Han populations were revealed. 

## 2. Methods and Materials

The investigators conducted the following work according to the Code of Ethics of the World Medical Association (Declaration of Helsinki) for experiments involving humans. The conception and implementation of this study were approved by the Ethics Committee of Fudan University (code: BE1806; date: 3 March 2018).

### 2.1. DNA Samples

A total of 870 male unrelated samples of Chinese Han ancestry were collected with appropriate informed consent, consisting of blood samples provided on Flinders Technology Associates (FTA) cards (Whatman International Ltd., Maidstone, UK). Among them, 565 lived in Zhaotong City, Yunnan Province, while the rest (*n* = 305) lived in Zibo City, Shandong Province, and the male ancestors of all donors had been living in the sampling sites for three generations. As Figure 1 shows, the geographic locations of Shandong and Yunnan Han populations are indicated in red, and the referenced populations are in azure. 

### 2.2. Typing of 24 Y Haplogroup-Determining SNP Markers and 27 Y-STR Loci

We considered the following 24 binary Y-SNP markers within the Y-SNP Pedigree Tagging System (Suzhou Microread Genetics, Suzhou, Jiangsu, China) [18]: E-M96, D-JST021355, N-M231, C-M130, O-P186, I-M170, IJ-M429, K-M9, QR-M45, G-M201, IJK-M522, D1a1a1-N1, D1a2a-P47, C2-M217, N1a1-M46, O1a-M119, O1b-M268, O1b2-M176, O2-M122, O2a1-KL1, O2a2-P201, O2a2b-P164, O2a2a1a2-M7, and O2a2b1a1-M117. The nomenclature definition of these makers was strictly adopted according to the Y Chromosome Consortium (YCC) [6,7], International Society of Genetic Genealogy (ISOGG, website: https://isogg.org/), Phylotree [19], and Y Chromosome Haplotype Reference Database (YHRD) [20,21]. We analyzed the following 27 Y-STR loci (AmpFLSTR™ Yfiler™ Plus Kit (Applied Biosystems, Foster City, CA, USA) [22]): DYS438, DYS392, DYS393, DYS437, DYS448, DYS390, DYS19, DYS385a/b, DYS391, DYS389I, YGATAH4, DYS533, DYS635, DYS389II, DYS456, DYS481, DYS439, DYS460, DYS458, DYS449, DYS570, DYS576, DYF387S1a/b, DYS627, and DYS518. 

Direct amplification (aka DNA extraction-free method) was performed using the GeneAmp^®^ PCR System 9700 (Thermo Fisher Scientific, Foster City, CA, USA) in accordance with the manufacturer’s instructions. For this, 1 mm^2^ of blood stain on an FTA card was cut using a Harris micro-punch (Sigma-Aldrich, Saint Louis, MO, USA) and then added into each reaction mix. In addition, 1 ng of male 9948 control DNA (Marligen Biosciences, Ijamsville, MD, USA) was amplified as the positive control in each batch. The amplification products were separated by capillary electrophoresis (CE) performed by a 3500xL Genetic Analyzer (Thermo Fisher Scientific). The typing result was determined by means of GeneMapper^®^ ID-X Software v1.4 (Thermo Fisher Scientific). For each sample, PCR–CE analysis was performed in two replicates to check that the genotyping was accurate. In addition, the variants detected in this study were confirmed by replicates of PCR–CE analysis.

The Y-SNP data in concert with the Y-STR data obtained were validated and submitted to the release R62 of the Y-chromosomal haplotype reference database (YHRD, https://yhrd.org). The assigned accession numbers of Shandong and Yunnan Han populations were YA004617 and YA004618, respectively.

### 2.3. Data Analysis

#### 2.3.1. Forensic Parameters and Statistical Analysis

The allele of the locus DYS389II was defined by subtracting that of DYS389I. Forensic parameters regarding haplotype information were calculated for the 27 Y-STR loci included in the AmpFLSTR™ Yfiler™ Plus Kit [22]. Haplotype diversity (HD) was estimated by the formula
HD=n(1−∑ pi2)n−1
in which, *n* denotes the sample size, and *p*_*i*_ represents the frequency of the ith haplotype [23]. Match probability (MP) and discrimination capacity (DC) were calculated as follows$$P=\sum^{\;}{p_{i}}^{2},\;\mathrm{and}\;DC=N_{d}/N_{t}$$
according to Shannon’s instruction [24]. In other words, MP was the sum of the squared of unique haplotypes’ frequencies, while DC denoted the ratio of the number of unique haplotypes (Nd) to the total number of haplotypes (*N*_*t*_). The allele frequency of each Y-STR locus was generated by direct counting. The estimation method of gene diversity (GD) was analogous to that used for HD, where *p*_*i*_ represents the frequency of the ith allele. To calculate these parameters, the “Basic.stats()” function included in the “heirfstat” package developed by Thierry et al. [25] was utilized. GD values were generated for all samples, as well as for several major Y-SNP haplogroups of the two Chinese Han populations. Chi-square test was performed using the “chisq.test()” function in R language (version 3.5.3, https://www.r-project.org/).

#### 2.3.2. Y-Chromosomal Haplogroup-Based Network Analysis

A median-joining (MJ) network based on 27 Y-STR haplotypes was constructed using Network 5.0.0.3 software (http://www.fluxus-engineering.com/sharenet.htm) [26], in order to uncover the phylogenetic relationships among the samples carrying special rare variants. The weight was set for all Y-STR loci according to their mutation rates [4]. 

#### 2.3.3. Phylogenetic Reconstruction on the Population Level

As reported, Nothnagel et al. analyzed genetic phylogeny based on 17 Y-STR haplotypic data from almost 38,000 Chinese male samples [27]. Till now, a huge number of 27 Y-STR haplotypes have been published worldwide. First and foremost, in order to verify the population genetic backgrounds of the collected samples, a phylogenetic relationship analysis based on 27 Y-STR haplotypes in AmpFLSTR™ Yfiler™ PCR Amplification Kit (Thermo Fisher Scientific) [28] was performed by comparing Shandong and Yunnan Han populations with other 13 referenced populations, including Guangxi Zhuang (*n* = 2314) [29], Hulun Buir Mongolian (*n* = 282) [30], Xinjiang Uyghur (*n* = 161), Xinjiang Kazak (*n* = 130) [31], Yunnan Yi (*n* = 66) [32], Tibet Chamdo Tibetan (*n* = 172), Tibet Shigatse Tibetan populations (*n* = 230) [33], and Han groups from Guangdong (*n* = 247) [34], Shenzhen (*n* = 136) [35], Hainan (*n* = 473) [36], Shanghai (*n* = 843) [37], Changzhou, Jiangsu (*n* = 1550) [38], and Jining, Shandong (*n* = 877) [39]. Software Arlequin (version: 3.5.2.2) [40] was utilized to conduct an analysis of molecular variance (AMOVA, [41]). In order to reconstruct the male genetic relationships among the 15 Chinese populations, the computed pairwise genetic distances (R_ST_) were then used to perform multidimensional scaling (MDS, [42]) by “MASS” package within R language. Haplotypes carrying intermediate alleles, copy number variations, and null alleles were removed. Significant values (*p*-values) of R_ST_ were evaluated with 10,000 permutations.

#### 2.3.4. Phylogenetic Reconstruction on the Individual Level

Haplotypes with intermediate alleles, copy number variations, and null alleles were removed from the individual-level phylogenetic reconstruction. Pairwise genetic distance (d-value) of 27 Y-STR loci was calculated using $=\overset{i}{\underset{n}{\sum }}{\left(a_{i}-b_{i}\right)}^{2}/2m_{i}/n$, according to Nei’s molecular evolutionary theory [43]. In the formula, *n* denoted the number of Y-STR loci, and i was the ith locus, while m_i_, a_i_, and b_i_ represented the mutation rate of the ith Y-STR loci and the genotyping information of two different individuals. As for the 24 Y-SNPs, all samples were covered. The calculation of genetic distance (D-value) using Y-SNP loci was adopted according to Nei
$$D_{ij}=\frac{1}{L}\overset{L}{\underset{k=1}{\sum }}d_{kij}$$
where i and j denote the ith and jth individuals, L is the number of Y-SNP loci, and d_kij_ equals 0 or 1 depending on whether their SNP alleles are identical or not [44]. Phylogeny was reconstructed and illustrated by a “complete” method with Hierarchical Clustering (“hclust”) function [45] of R language. The MDS plots of 27 Y-STRs and 24 Y-SNPs on individual level were illustrated as described above.

## 3. Result and Discussion

### 3.1. MDS and AMOVA

With the aim to verify the sampling representativeness of Shandong and Yunnan Han populations and to reveal the genetic backgrounds of the two populations located in southern and northern parts of China, population data of other 13 representative Chinese populations (Figure 1) were selected, and the population structure was reconstructed. AMOVA analysis (Appendix A) based on all 27 Y-STR markers was conducted for the 15 populations and visualized in an MDS plot (Figure 2).

After Bonferroni correction, the significant difference was set to 0.05/105 ≈ 0.0005. Insignificant differences, referred to *p*-values above 0.0005 which are not indicated in bold in Appendix A, were only observed for three southern Chinese Han populations from Guangdong, Shenzhen, and Yunnan, which indicated their close consanguinity to each other. The Altaic-speaking populations were all significantly distant from Sino-Tibetan-speaking groups. Additionally, explicit differences could be found among some Han populations with ancestry from northern, central, and southern Chinese Han divisions.

In Figure 2, two MDS plots ware illustrated to explain the genetic landscape of various Chinese ethnic groups (initial stress = 0.0573), as well as the genetic make-up of Han groups (initial stress = 0.0824). Both plots reached a good quality of configuration. The results for the Altaic- and Sino-Tibetan-speaking groups complied with the AMOVA results, except for Yunnan Yi population, which may be caused by its small sample size. The distribution pattern on the abscissa axis explained the divergence between Chinese Han and Tibetan groups. As is known to all, genetics is strongly correlated with linguistics, as language carries cultural information. Inter-disciplinary efforts by archaeology, genetics, and linguistics help provide insights into historical human evolution [46,47]. The latest linguistic finding by Zhang et al. showed that the Chinese Han and Tibetan subgroups originated from a Sino-Tibetan language family which diverged about 4200–7800 years BP (before present), with an average value of 5900 years BP [48]. The phylogenetic evidence could be traced back to the late Neolithic.

Specific to the Chinese Han populations cluster, another MDS plot focused on the construction of the inner structure of Han population to dissect subtle population relationships. The pattern matched the substructure of Han Chinese described in a previous study [9]. The eight populations were divided into four clusters which matched their geographic locations approximately: Shandong and Jining (top), Shanghai (bottom left), Hainan (right), and Changzhou, Jiangsu, Guangdong, Shenzhen, and Yunnan (middle). Though geographically close, the Han population from Shanghai was not genetically close to those from Changzhou and Jiangsu, which may be related to the persistent Chinese migration to the metropolitan Shanghai [49]. In addition, the Changzhou Han population was genetically close to three Chinese Southern Han populations (Guangzhou Han, Shenzhen Han, and Yunnan Han) rather than to other central or northern Han populations, which indicated that the major component of male Changzhou Han population was from southern China. Primarily, the reconstructed genetic structure demonstrated the different genetic backgrounds of the two Han populations analyzed, which were genetically similar to other geographically close populations independently. The samples enrolled had a high degree of population representativeness.

### 3.2. Y-SNP Allelic Frequencies and Haplogroup Distribution

For both populations, 24 Y-SNPs were analyzed, and 18 Y chromosome haplogroups were defined (D, D1a1a1, C, C2, IJ, K, QR, N, N1a1, O1a, O1b, O1b2, O2, O2a1, O2a2, O2a2a1a2, O2a2b, and O2a2b1a1). The distribution of haplogroups within the Shandong and Yunnan Han populations are displayed in Figure 3.

In order to demonstrate substructure differences, we analyzed the Y chromosomal sub-haplogroup characteristics between in two Han populations. Y-SNP allelic frequencies of the two populations were compared (Figure 3A) to figure out the primary differences. Four Y-SNP loci with no derived allele (E-M96, I-M170, G-M201, and D1a2a-P47) are not shown. Significant differences in 5 of the 24 Y-SNP allelic frequencies (*p* = 0.05/20 = 0.0025) were discovered. In detail, the Yunnan Han population showed much higher frequencies in O1a-M119 and O1b-M268, but lower frequencies in O2-M112, O2a2-P201, and O2a2b-P164, which was also reflected by the disparity in haplogroup distribution (Figure 3B).

Over 70% of the samples of both populations were from haplogroup O [50]. The most frequent Y lineages in Yunnan Han samples were found to be O1a (16.3%), O1b (14.9%), O2a2b1a1 (14.9%), O2a1 (12.7%), O2a2b (11.5%) and C2 (8.1%), while that in Shandong Han samples were O2a2b (23.3%), O2a1(18.4%), O2a2b1a1(14.1%), C2(12.8%), and O1b (6.6%). The haplogroup distributions were consistent with the results of a previous study [51], which showed that the four major Y chromosome haplogroups in East Asian males are D-M174, C-M130 (not including C5-M356), N-M231, and O-M175 and that haplogroup O is found in the majority of Chinese males.

The haplogroup C is typical of the residents of Eurasian temperate steppe and can be found in northern Han populations as well. The high proportion of the C2 haplogroup in northern Han (Shandong Han) might be related to nomad incursions into the Central Plain in history [52]. The differences in haplogroups O1 and O2 of Shandong and Yunnan Han populations might be the result of the initial founder effect of the early large-scale north migration and of geographical isolation [53]. The O2 haplogroup was probably dominant among northward migrants after population expansion in southern China, causing a higher proportion of the O2 haplogroup in Shandong Han.

Four haplogroups—O1a, O1b, O2a1, and O2a2b, were characterized by dominant but different distributions in the two populations. The O1a haplogroup is mainly distributed in southern China, Malaysia, Vietnam, and Indonesia males. The O1b haplogroup is unique to modern Eastern Eurasian populations. Previous findings indicated that the proportion of the O1 haplogroup is significantly higher in southern China compared with northern China [54], which was also confirmed in this research. In contrast, the two haplogroups O2a1 and O2a2b, which are also dominant in East Asian populations [55], had relatively higher proportions in the Shandong Han population.

### 3.3. Y-STR Allele Variants

In total, 55 null alleles were observed at 14 different loci (DYS448, DYS390, DYS391, DYS456, DYS481, DYS460, DYS458, DYS449, DYS570, DYS576, DYS627, DYS518, and DYF387S1a/b). Furthermore, 71 micro-variants were found in 69 samples, including 8 single-copy loci, i.e., DYS518 (36.2, 37.2, 38.2 and 39.2), DYS627 (17.2, 18.2 and 21.2), DYS448 (18.1, 18.2, 19.2 and 20.2), DYS438 (10.1), DYS458 (14.1 and 15.1), DYS570 (19.2), DYS449 (30.1 and 30.2), and DYS481 (23.2), and two multi-copy loci, i.e., DYF387S1a/b (35.3/35.3, 36.3/40, 35.3/37.3, 37.3/39, 37.3/37.3, and 36.3/38.3) and DYS385a/b (13.1/13.1, 12.1/13.1, 12/16.1, and 12/17.2). We found 10 copy number variations (CNVs): DYS518 (37/38), DYS390 (23/24), DYS437 (15/16), DYS389 I (12/13), DYS576 (19/21), DYS456 (15/16 and 17/18), DYS385 (13/19/20), and DYF387S1 (34/35/36) in other 10 samples. After searching in the YHRD database and other referenced publications [29,34,35,36,37,56,57,58,59,60,61,62,63,64], DYS570 (19.2), DYS481 (23.2), DYF387S1 (35.3/35.3, 36.3/40, 35.3/37.3, 37.3/39 and 36.3/38.3), DYS385 (12.1,13.1), DYS576 (19,21), and DYS456 (15,16) were reported for the first time. All samples with variants are co-listed with their Y haplogroup affiliations in Appendix A.

Interestingly, among the 69 samples with micro-variants, 29 carried “.2” mutation at DYS518, which were all found to be descents of QR haplogroup ancestors, accounting for 64.4% of individuals assigned to the QR-M45 haplogroup. Albeit it has been concluded that the QR haplogroup is not a major haplogroup of the East Asian population [65], 92.2% of the reported “DYS518~.2” mutations were found in Chinese samples in the YHRD database. Additionally, Lang et al. found a relationship between the “DYS518~.2” alleles and the haplogroup Q [66]. In order to define the evolutionary history of the “DYS518~.2” allele, the median-joining network was utilized to construct the inner structure of the QR-M45 haplogroup (Figure 4). The ancestral structure is indicated by the red torso of the network, with unmutated event at DYS518, indicating the allele 37 was possibly the ancestral allele. All nodes of the samples are linked so to form two independent clusters. In addition, two nodes for allele 37.2 and other two nodes for the unmutated allele 38 are located closely at the joint of two clusters. The closeness of these samples indicated that “DYS518~.2” alleles likely derived from the mutated allele 37.2, which might be characteristic for the QR haplogroup in Chinese populations. However, the underlying evolutionary pathway leading to the shift of unmutated allele 38 to mutated allele 37.2 remains unclear. In order to explain this observation, more samples from the QR haplogroup should be collected and profiled utilizing massive parallel sequencing technology. A higher resolution definition of Y-DNA paragroups will provide insights for a comprehensive knowledge of Y-STR haplotype evolution in ancient major haplogroups.

### 3.4. Distribution Spectrums of Y-STR Alleles within C2, O1, and O2 Haplogroups in Shandong and Yunnan Han Populations

To uncover the varying patterns of Y-STRs within haplogroups from different populations, three major haplogroups, i.e., C2, O1, and O2 were selected, to which the majority of the Shandong and Yunnan Han samples belonged (Appendix A).

Significant differences of allelic frequencies were observed at DYS627 within all three haplogroups, which was also the only difference observed in both O2 and C2. In O1, however, significant differences could also be observed in single-copy loci DYS481, DYS389I, DYS389II, and DYS570 and in the multi-copy locus DYS385a/b. Albeit the varying patterns of most Y-STR loci showed close correlation with Y haplogroups instead of populations, significant differences at some Y-STR loci within the identical major Y haplogroups may reflect their different ancestral sources. In general, the regularity of a varying pattern demonstrated that the Y-STR gene pool remained stable, regardless of the different haplotypes in Yunnan and Shandong Han populations or the big differences in geographic and cultural definitions. Further, the different patterns for different major haplogroups revealed the primary superiority of Y-SNP haplogroups in classifying male groups, dissecting population structure, and exploring population migration. In forensic practices, especially for the Chinese Y-STR haplotype database which includes tens of millions of Y-STR haplotypes, Y-SNP could play a critical role for pedigree discrimination, as well as biogeographic inference.

### 3.5. Forensic Parameters

The GDs of all 27 Y-STR markers were calculated both for the three major haplogroups (C2, O1, and O2) of the two populations separately and for the total population (Appendix A, Figure 5). For the total population, it could be found that albeit gene diversity of most Y-STR loci was high (>0.5), in some cases, it was low, such as for DYS438 (0.2664), DYS437 (0.1777), DYS391 (0.3769), DYS392 (0.3847), and DYS393 (0.3339). Furthermore, some Y-STRs in different Y haplogroups presented different gene diversities, such as DYS437, which had an extremely low gene diversity in both C2 and O1 haplogroups but very high values in the O2 haplogroup. In addition, although the gene diversity in DYS533 was low in the C2 haplogroup in Shandong, it showed a very high value in the same haplogroup in Yunnan. The same was observed for DYS456 in the O1 haplogroup. This indicated the presence of sub-structures within different haplogroups from different regions.

Moreover, all Y-STR markers were used to analyze the classic forensic parameters in the two populations. Among all 870 samples, 864 haplotypes were unique (Appendix A). There were four haplotypes with two repetitions, and one with three repetitions. According to the different panels within the Yfiler and Yfiler Plus amplification systems (17 and 27 Y-STR loci), standard forensic parameters (HD, DC, and MP) in 305 Shandong Han samples and 565 Yunnan Han samples were separately calculated. Also, these three parameters were estimated among the K, O2, O2a2, O2a2b, and O2a2b1a1 haplogroups from the two Han populations, as only repetitive Y-STR haplotypes could be observed within both the major and the in-depth clades of the K haplogroup (Table 1, Figure 6). Repetitive Y-STR haplotypes mean that the Y-STR haplotypes of two different males are identical. If all Y-STR haplotypes (number = *n*) in one population are different from each other, the forensic parameters HD, MP, and DC would be equal to 1, 1⁄n, and 1, respectively, and it would not be worth comparing them.

The HD and DC values of Yunnan Han population were comparatively greater than those of Shandong Han population, though they were high for both populations. The MP value of Yunnan Han was smaller. The varying patterns of the forensic parameters indicated that the 10 new Y-STR loci incorporated within the Yfiler^™^ Plus kit helped to significantly increase the haplotype diversity and discrimination capacity but decreased the match probability in populations at various scales. For the sub-population composed of samples assigned to higher resolution haplogroups, MP was higher, which conformed to the common knowledge that one Y-STR haplotype would be liable to match those from the same Y-SNP haplogroups. Thus, for samples assigned to the identical high-resolution haplogroup, more Y-STRs are required to identify unrelated males.

### 3.6. Phylogenetic Reconstruction on the Individual Level

In Figure 7, it was explicit that the phylogenetic tree could basically cluster the Y-STR haplotypes from the same Y-SNP sub-haplogroups. However, there were also several disparities, since a total of 51 samples (proportion = 6.4%) were observed in various unfitting regions of the phylogenetic tree. Of these, 28 were located inside the major haplogroup, while the rest crossed the major haplogroups. In addition, though the individuals assigned to the identical Y chromosomal haplogroup were clustered, the phylogeny based on 27 Y-STR loci was significantly different from the Y-DNA tree reported [19]. Some samples from the same haplogroup located in several clusters. Clearly, albeit the limited number of Y-SNP loci selected and Y chromosomal haplogroups, Y-STRs combined with Y-SNPs would help increase the discriminability of male pedigrees in the simulated Y chromosome database.

Significantly, the structures observed in the MDS plots were different. Figure 8A demonstrates that the 27 Y-STR loci were not able to distinguish Han males in Shandong and Yunnan populations (Figure 8A), because most individuals clustered together. However, the 24 Y-SNPs showed potential to clearly classify male individuals as various haplogroups (Figure 8B). In addition, individuals assigned to the C-M130 and D-JST021355 haplogroups were all from Yunnan Han population, while those belonging to the K-M9 haplogroup were all from Shandong Han population, indicating the possible bio-geographic discrimination ability of Y-DNA-haplogroup-determining SNPs.

## 4. Conclusions

In summary, Shandong and Yunnan Han populations, the representatives of northern and southern Chinese Han, were focused on to investigate their genetic backgrounds via 27 commonly used Y-STRs and 24 East-Asian-haplogroup-determining Y-SNPs. Among the 870 samples, 864 haplotypes were unique. The observed Y-STR allele variants including null alleles, intermediate alleles, and CNVs were summarized. Of these, “DYS518~.2” alleles were all found within QR haplogroup individuals, and a network was constructed to characterize the evolutionary pathway of this kind of variant. Primarily, the forensic parameters (GD, HD, DC, and MP) within different Y chromosomal haplogroups furnished the evidence that the co-application of Y-STR and Y-SNP analysis would provide more informative characteristics of various populations. A phylogenetic reconstruction on the individual level further explained that Y-STRs combined with Y-SNPs would help increase the discriminability of male pedigrees using a Y chromosome database. This study sheds light on basic genetic backgrounds utilizing both Y-STR and Y-SNP loci, showing their usefulness for uncovering detailed population differences. More importantly, this tentative study will likely help to build a Y-SNP databank to promote Chinese male pedigree discriminability.

## Figures and Tables

**Figure 1 genes-11-00743-f001:**
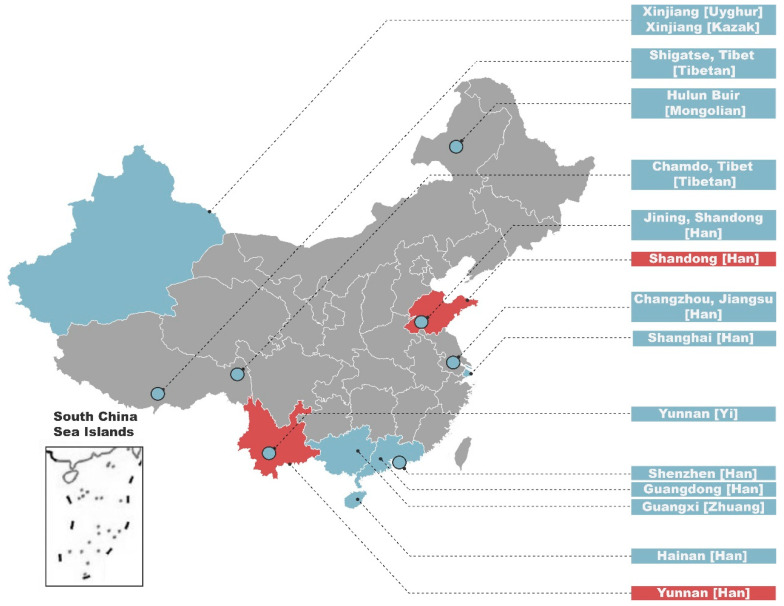
Geographic localizations of the 15 Chinese populations analyzed or referenced. The two populations reported in this study are indicated in red, whereas the other 13 referenced populations in the analysis of molecular variance (AMOVA) are indicated in azure.

**Figure 2 genes-11-00743-f002:**
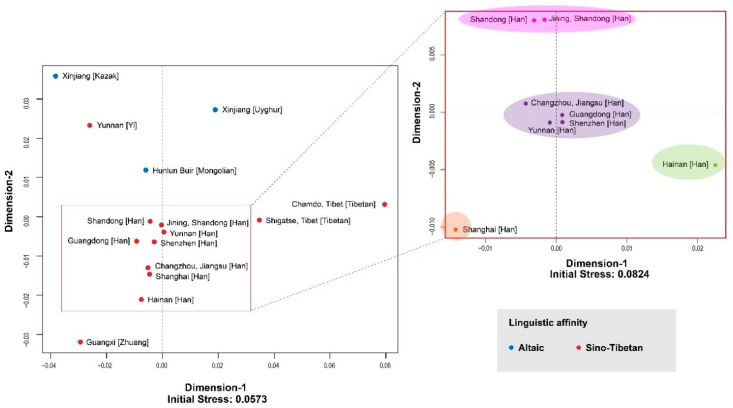
Multidimensional scaling (MDS) plots of 15 Chinese Han populations based on pairwise genetic distances (R_ST_). In the left plot, blue dots indicate Altaic-speaking groups, and red dots indicate Sino-Tibetan-speaking populations. In the right plot, Han populations are divided into four clusters which are labeled with various light colors.

**Figure 3 genes-11-00743-f003:**
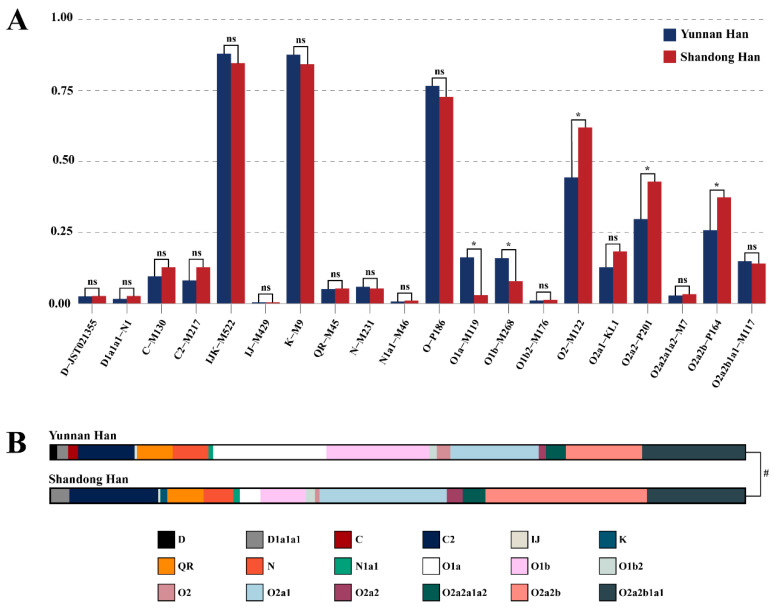
Allelic frequency and haplogroup distribution determined for 24 Y chromosomal single-nucleotide polymorphisms (Y-SNPs) in Yunnan and Shandong Han populations. (**A**) Frequencies for 20 Y-SNP loci with derived allele. Yunnan Han population is represented by blue bars, while Shandong Han population is represented by red bars. “*” and “ns” denote if the allelic frequency comparison is significant or not, respectively. (**B**) Distribution of Y chromosome haplogroups. Different haplogroups are presented with different colors. The upper bar is the haplogroup distribution of Yunnan Han, and the lower one is that of Shandong Han. “#” denotes a significant difference in haplogroup distribution.

**Figure 4 genes-11-00743-f004:**
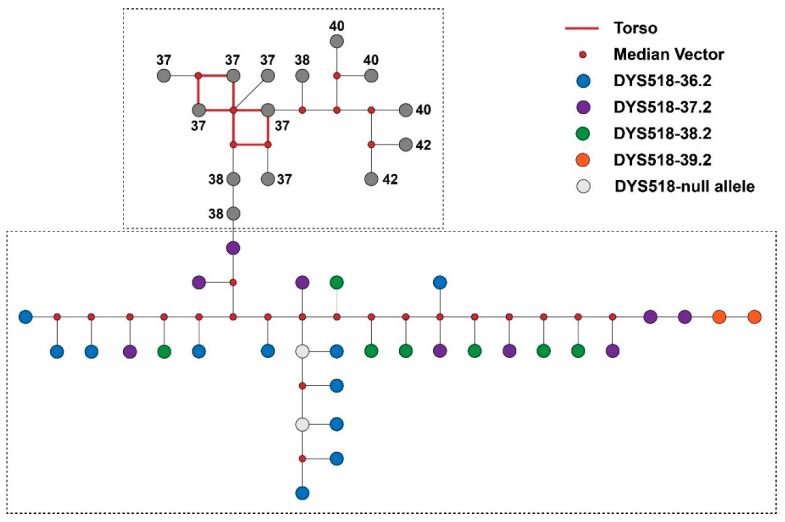
Median-joining network of Y chromosomal short tandem repeats (Y-STR) haplotypes of samples of the haplogroup QR-M45. Blue nodes indicate samples with the DYS518~36.2 variant, purple nodes indicate samples with the DYS518~37.2 variant, green nodes indicate samples with the DYS518~38.2 variant, and orange nodes indicate samples with the DYS518~39.2 variant.

**Figure 5 genes-11-00743-f005:**
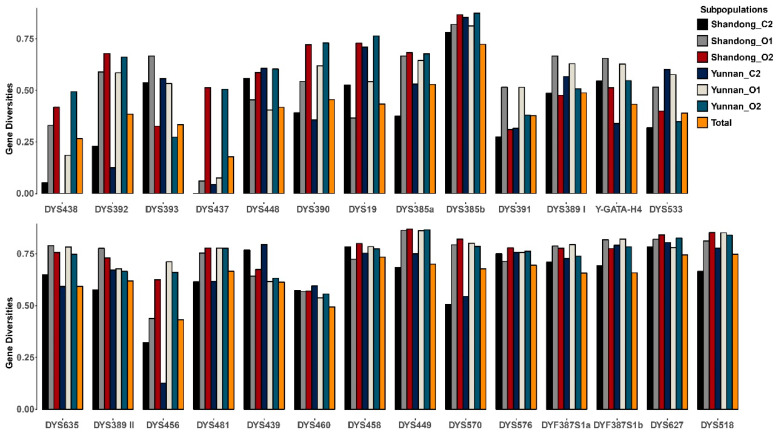
Gene diversity values for the 27 Y-STR markers in different major Y-chromosomal haplogroups. The bar colors indicate the gene diversity (GD) values in different major Y haplogroups (C2, O1, and O2) of each population and in all 870 samples. The order of Y-STR markers is arranged along with Y-STR mutability (from low to high) [4].

**Figure 6 genes-11-00743-f006:**
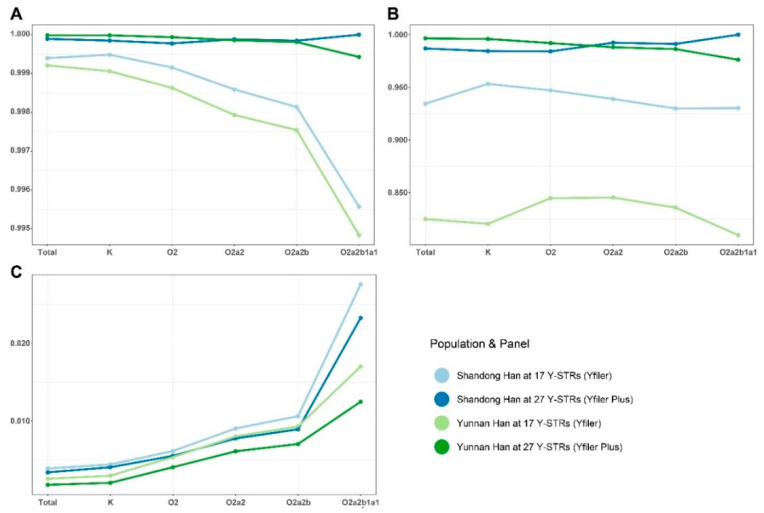
Varying patterns of haplotype diversity (HD), discrimination capacity (DC), and match probability (MP), calculated for different Y-STR panels, in the whole groups of Shandong and Yunnan Han as well as in the sub-populations defined by Y-SNPs. (**A**) HD values. (**B**) DC values. (**C**) MP values. Different colors denote different populations for the 17- or 27-Y-STR panel.

**Figure 7 genes-11-00743-f007:**
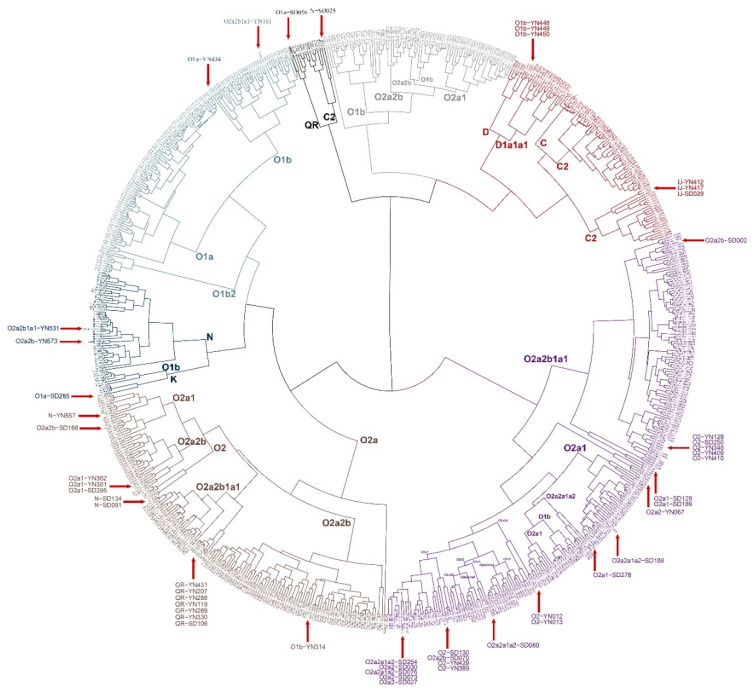
Phylogeny tree based on the 27-Y-STR panel. All samples are marked with the measured Y haplogroup, where “YN” and “SD” are abbreviations for Yunnan and Shandong, respectively. Different colors refer to various clusters. Red arrows denote those near-match Y-STR haplotypes which were classified into other different or proximate Y-SNP clades.

**Figure 8 genes-11-00743-f008:**
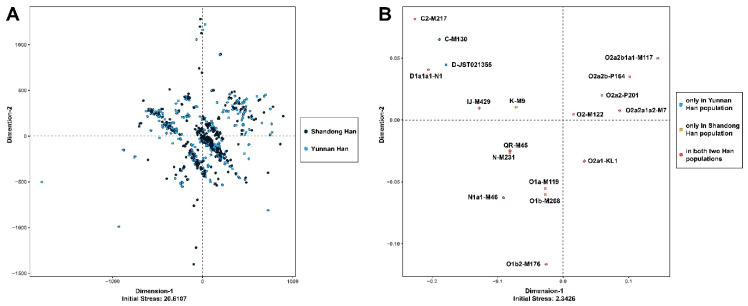
MDS plots on the individual level using 27 Y-STR loci and 24 Y-SNPs. (**A**) Plot for 27 Y-STR loci. Male individuals from Shandong Han population are labeled in navy blue, while those from Yunnan Han population are in light blue. (**B**) Plot for 24 Y-SNPs. Blue denotes male individuals from Yunnan Han population; yellow denotes those from Shandong Han population; jacinth denotes male individuals from both populations.

**Table 1 genes-11-00743-t001:** Standard forensic parameters based on 17 or 27 Y-STR loci for the whole groups of Shandong and Yunnan Han, as well as for the sub-populations defined by Y-SNPs.

Population	Panel	HD	DC	MP	Unique Haplotypes	Sample Size
Yunnan Han						
Total	Yfiler-17	0.99921	0.82478	0.00256	466	565
	YfilerPlus-27	0.99999	0.99646	0.00178	563	565
K	Yfiler-17	0.99906	0.82020	0.00296	406	495
	YfilerPlus-27	0.99998	0.99596	0.00204	493	495
O2	Yfiler-17	0.99863	0.84462	0.00535	212	251
	YfilerPlus-27	0.99994	0.99203	0.00405	249	251
O2a2	Yfiler-17	0.99793	0.84524	0.00801	142	168
	YfilerPlus-27	0.99986	0.98810	0.00609	166	168
O2a2b	Yfiler-17	0.99754	0.83562	0.00929	122	146
	YfilerPlus-27	0.99981	0.98630	0.00704	144	146
O2a2b1a1	Yfiler-17	0.99484	0.80952	0.01701	68	84
	YfilerPlus-27	0.99943	0.97619	0.01247	82	84
Shandong Han						
Total	Yfiler-17	0.99940	0.93443	0.00388	285	305
	YfilerPlus-27	0.99989	0.98689	0.00339	301	305
K	Yfiler-17	0.99948	0.95331	0.00441	245	257
	YfilerPlus-27	0.99985	0.98444	0.00404	253	257
O2	Yfiler-17	0.99916	0.94709	0.00613	179	189
	YfilerPlus-27	0.99977	0.98413	0.00551	186	189
O2a2	Yfiler-17	0.99859	0.93893	0.00903	123	131
	YfilerPlus-27	0.99988	0.99237	0.00775	130	131
O2a2b	Yfiler-17	0.99814	0.92982	0.01062	106	114
	YfilerPlus-27	0.99984	0.99123	0.00893	113	114
O2a2b1a1	Yfiler-17	0.99557	0.93023	0.02758	40	43
	YfilerPlus-27	1.00000	1.00000	0.02326	43	43

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
