# Peer review of "Genetic Reconstruction and Forensic Analysis of Chinese Shandong and Yunnan Han Populations by Co-Analyzing Y Chromosomal STRs and SNPs"

_genes, 2020, doi:10.3390/genes11070743_

Round 1

Reviewer 1 Report

This paper compares Y STR allelic variation in Chinese population groups with haplotypes and haplogroups defined by Y SNPs.  The population genetic analyses are rigorous (if not complete) and there are some interesting findings, probably the most significant being the origins of a .2 micro-variant at the DYS518 locus in the QR haplogroup.  However, the promise in the title was not fully realised as there was incomplete investigation of any correlation between Y STR haplotypes and Y SNP haplotypes/haplogroups.  Conventional wisdom suggests that STRs are not as useful as SNPs for population structure because of their higher mutation rates which tend to provide a lot of noise in any population signal.

What is missing is a plot showing correlation (or not) between Y SNP haplotypes or haplogroups and Y STR haplotypes.  This could be, for example, a plot similar to Figures S1, S2 and S3 where the different colours refer to the 18 identified haplogroups (or perhaps the largest haplogroups: O1a, O2a1, O1b, O2a2b1a1, O2a2b, C2), rather than Shandong and Yunnan.  There may be better ways to explore such a correlation and the authors may be able to improve on my suggestion.

In Figure 2, I think the colour coding by population division just leads to confirmation bias.  The Guangdong, Shenzhen and Yunnan populations in the Southern Han group and the Changzhou, Jiangsu population in the Central Han group are more similar to each other than they are to other populations in the same group.  Figure 2 suggests to me that there are four population groups:

  1. Shandong and Jining (top)
  2. Shanghai (bottom left)
  3. Hainan (right)
  4. Changzhou, Jiangsu, Guangdong, Shenzhen and Yunnan (middle)

I would also like to see an MDS plot similar to Figure 2 based on pairwise differences between Y SNP haplotypes.  If there is any population structure amongst northern, central and southern Han, it should also be reflected in Y SNPs. 

Lines 90-95: Samples were collected from “unrelated” male donors “all within three generations”.  What does this mean?  If the donors were unrelated, how could they be within three generations?

Lines 341-342: What are “repetitive STR haplotypes”?

Some minor editing for English required.

Reviewer 2 Report

Dear Authors, Most respectfully I submit that this is another paper around interesting Chinese populations using Y SNPs and STRs. Significant number of samples have been profiled for 24 Y SNPs and 27 Y STRs using standard kits and methods. The statistical evaluation of the data has been done in a standard fashion and some of the data has been shown in interesting manner. There are good tables and figures which clarify the understanding of the data. Y STRs are important tools for forensic and evolutionary studies as are Y SNPs. This is a novel study in terms of haplogrouping & haplotyping of same samples and defining the haplotypes within specific haplogroups. There are novel findings though these need to be further confirmed. Some of the work around ancestral nature of alleles has been attempted which is hypothetical in nature. I agree that a much larger dataset would be required within specific haplogroups in order to get a better results. For mating of the paper, referencing and flow of thought process is of a good standard. Thanks & Best Regards

Reviewer 3 Report

The paper “Genetic reconstruction and forensic analysis of Chinese Shandong and Yunnan Han populations by co-analyzing Y chromosomal STRs and SNPs” provided data of Y-STR haplotypes for 27 markers and Y-SNP haplogroups defined by 24 loci in Shandong Han (n = 305) and Yunnan Han (n = 565) populations.  The observed genetic backgrounds would be useful for uncovering detailed population differences and demonstrated the supplementary role of Y-SNPs in population differentiation.

The manuscript is well written, the introduction provides a good background of the topic and the methods are appropriate for the study. The paper is interesting and worthy of being published, after clarifying the following minor points:

-Row 121: replicates are referred to PCR - CE analysis or CE alone?

-Row 177: The sentence “insignificant differences were only observed among 3 southern Chinese Han populations from Guangdong, Shenzhen and Yunnan” is unclear. The p-values are below the 0.0005  (not indicated in bold in table S2?) or above?

-Row 258: the variants detected in the study have been confirmed by replicates of PCR?

-Table S1 and Table S3: please be careful of the use of points or commas.

-Table S2: Please note the micro-variants with a .1 .2 etc… as correctly reported in the manuscript rather than with a comma (i.e. 19,2 in the first row of table S2).

Round 2

Reviewer 1 Report

Thank you to the authors for accommodating my suggestions.

The new Figure 7 is a good way to compare phylogenies based on Y SNPs and Y STRs. 

Although there are no Y SNP data for other northern, central or southern Han populations, it would still be possible to construct an MDS plot for just the 305 Shandong Han individuals and the 565 Yunnan Han individuals for both the 24 Y SNPs and 27 Y STRs.  It would be interesting to see if they look similar.

Some minor editing for English is still required.
